

# Differentially expressed genes in orbital adipose/connective tissue of thyroid-associated orbitopathy

Yan Wang[1], Yanqiu Liu[1], Jiping Cai[1], Tianyi Zong[1], Ziyin Zhang[1], Tianhua Xie[1], Tong Mu[1], Meili Wu[2], Qian Yang[1,2], Yangningzhi Wang[1], Xiaolu Wang[2] and Yong Yao[1]

[1] Department of Ophthalmology, The Affiliated Wuxi People's Hospital of Nanjing Medical University, Wuxi, China
[2] Center of Clinical Research, The Affiliated Wuxi People's Hospital of Nanjing Medical University, Wuxi, China

## ABSTRACT

**Background.** Thyroid-associated orbitopathy (TAO) is a disease associated with autoimmune thyroid disorders and it can lead to proptosis, diplopia, and vision-threatening compressive optic neuropathy. To comprehensively understand the molecular mechanisms underlying orbital adipogenesis in TAO, we characterize the intrinsic molecular properties of orbital adipose/connective tissue from patients with TAO and control individuals.

**Methods.** RNA sequencing analysis (RNA-seq) was performed to measure the gene expression of orbital adipose/connective tissues of TAO patients. Differentially expressed genes (DEGs) were detected and analyzed through Gene Ontology (GO), Kyoto Encyclopedia of Genes and Genomes (KEGG) analysis, and Gene Set Enrichment Analysis (GSEA). The protein–protein interaction (PPI) network was constructed using the STRING database, and hub genes were identified by the Cytoscape plug-in, cytoHubba. We validated several top DEGs through quantitative real-time polymerase chain reaction (qRT–PCR).

**Results.** We identified 183 DEGs in adipose tissue between TAO patients ($n = 3$) and control patients ($n = 3$) through RNA sequencing, including 114 upregulated genes and 69 downregulated genes. The PPI network of these DEGs had 202 nodes and 743 edges. PCR-based validation results of orbital adipose tissue showed multiple top-ranked genes in TAO patients ($n = 4$) are immune and inflammatory response genes compared with the control individual ($n = 4$). They include ceruloplasmin isoform x3 (CP), alkaline tissue-nonspecific isozyme isoform x1 (ALPL), and angiotensinogen (AGT), which were overrepresented by 2.27- to 6.40-fold. Meanwhile, protein mab-21-like 1 (MAB21L1), phosphoinositide 3-kinase gamma-subunit (PIK3C2G), and clavesin-2 (CLVS2) decreased by 2.6% to 32.8%. R-spondin 1 (RSPO1), which is related to oogonia differentiation and developmental angiogenesis, was significantly downregulated in the orbital muscle tissues of patients with TAO compared with the control groups ($P = 0.024$).

**Conclusions.** Our results suggest that there are genetic differences in orbital adipose-connective tissues derived from TAO patients. The upregulation of the inflammatory response in orbital fat of TAO may be consistent with the clinical phenotype like eyelid edema, exophthalmos, and excess tearing. Downregulation of MAB21L1, PIK3C2G,

Corresponding authors
Xiaolu Wang, xlwang@njmu.edu.cn
Yong Yao, yongyao@njmu.edu.cn

and CLVS2 in TAO tissue demonstrates dysregulation of differentiation, oxidative stress, and developmental pathways.

## INTRODUCTION

Thyroid-associated orbitopathy (TAO), also called Graves' ophthalmopathy, is a category of autoimmune diseases associated with thyroid dysfunction (*Bahn, 2010*). A prominent feature of TAO is the expansion of orbital tissue, comprising both extraocular adipose and muscle tissues (*Garrity & Bahn, 2006*). The swollen soft tissues are the result of the accumulation of nonsulfated glycosaminoglycan, inflammation, hyaluronan, and the activation of local fibroblasts (*Berchner-Pfannschmidt et al., 2016*). If left untreated, the expansion of orbital tissue can result in orbital congestion, significant exophthalmos, compressive neuropathy, and even lead to vision loss causing a serious decline in quality of life (*Wang et al., 2021*). In the last several decades, rehabilitative orbital decompression surgery has been the standard treatment for the stable stage of TAO. This surgical approach aims to mitigate proptosis, alleviate orbital congestion, and enhance the aesthetic appearance of the orbital region. Consequently, it serves as a means to ameliorate the quality of life for individuals afflicted with TAO (*Bartalena, 2013*).

The activation of orbital fibroblasts plays a key role in the immune process of TAO pathogenesis (*Naik et al., 2010*). Under pathological conditions, orbital fibroblasts will express functional molecules, such as thyrotropin receptor, the receptor of insulin-like growth factor, and CD40, and continue to differentiate into adipocytes and myofibroblasts closely related to disease progression. Most of the current studies focus on isolating and establishing primary orbital fibroblasts and conducting further immune research related to various pathological mechanisms of TAO (*Hammond et al., 2021*; *Jang et al., 2019*). However, limited research has been conducted concerning the direct detection of gene expression within the orbital adipose/connective tissue of TAO patients utilizing high-throughput sequencing methods. This issue emphasizes the importance of comprehending the underlying mechanism(s) of orbital adipogenesis to identify therapeutic approaches for the prevention or treatment of TAO.

The transcriptome refers to the sum of all RNA transcripts for a specific tissue or cell in a certain developmental state or functional condition, including messenger RNA (mRNA), noncoding RNAs, and small RNAs. Screening the specific genes that play a key role in disease among many differentially expressed genes (DEGs) has become a key research goal (*Chen et al., 2021*). Bioinformatics analysis based on gene expression profiles may screen hub genes and regulatory pathways, which play an important role in the early diagnosis of TAO and the establishment of early warning mechanisms (*Kim et al., 2021*).

In this study, DEGs were identified based on high-throughput RNA sequencing data of tissues from TAO and control subjects to explore the pathogenesis of TAO. Then, Gene Ontology (GO), Kyoto Encyclopedia of Genes and Genomes (KEGG), and Gene Set Enrichment Analysis (GSEA) pathway analyses were obtained to predict the functions of these DEGs. The expression patterns of some DEGs were confirmed by qRT-PCR.

## MATERIALS & METHODS

### Subjects and tissue samples

All human studies were conducted according to the Declaration of Helsinki principles and were approved by the Ethics Committee of the Affiliated Wuxi People's Hospital of Nanjing Medical University (identifier, KY23013). We collected human orbital adipose/connective tissues from 43 to 80-year-old patients with TAO undergoing routine resection of prolapsed orbital fat in the Department of Ophthalmology, the Affiliated Wuxi People's Hospital of Nanjing Medical University, from July 2021 to August 2022. The demographics of the patients are presented in Table 1 and Fig. S1. All TAO patients included in this study were diagnosed according to Bartley's criteria, and tissues of control individuals obtained in plastic surgery were collected as control samples. All patients provided written informed consent.

### Bulk RNA sequencing analysis (RNA-Seq)

The total RNA in tissues were extracted. To ensure the quality of the samples for transcriptome sequencing, the concentration and integrity of RNA samples were checked using a Nanodrop ND-2000 spectrophotometer and an Agilent Bioanalyzer 2100/4200, respectively. The qualified RNA samples were used for mRNA preparation and cDNA library construction. After library construction, the qualified libraries were sequenced using the Illumina NovaSeq 6000 using PE150 mode. Following an extracting and filtering quality control, we obtained high-quality, cleaned reads, and a follow-up analysis was then conducted (Table S1). All experiments were repeated three times with biological replicates. The statistical power of this experimental design, calculated in RNASeqPower is 0.96, based on a sequencing depth of 6 GB, CV of 0.4. We have uploaded the RNA-seq into the NCBI, the NCBI accession number is PRJNA971380.

### DEGs and differential alternative splicing (DAS) analysis

We used FeatureCount (version 2.0.2) (*Liao, Smyth & Shi, 2014*) to quantify transcripts at the gene level. Differential expression analyses were performed with edgeR (version 3.3.3) according to the criteria of $|\log 2 (FC)| > 1$ and $P$ value $< 0.05$.

Alternative splicing (AS) is the process by which different splice sites in precursor messenger RNA are selected to generate multiple mRNA isoforms, so AS is an important mechanism in creating proteome diversity and regulating gene expression in different tissues and developmental stages. To identify the number of different splicing events in TAO patients and controls, the software rMATS (version 4.0.2) was used (*Shen et al., 2014*), a new statistical method for robust and flexible detection of differential AS from replicate RNA-Seq data. Five main alternative splicing events, A3SS, A5SS, MXE, RI, and

**Table 1 Characteristics of TAO and control patients undergoing study.**

|  | Control ($n = 6$) | TAO ($n = 5$) | *P* value |
|---|---|---|---|
| **Age (years)** | $32.67 \pm 13.57$ | $59.20 \pm 13.29$ | 0.010 |
| **Male (n, %)** | 1 (16.67%) | 3 (60%) | 0.137 |
| **Disease (n, %)** |  |  |  |
| TAO | — | 5 (100%) |  |
| Blepharochalasis | 2 (33.33%) | — |  |
| Adipositas palpebrae | 1 (16.67%) | — |  |
| Exotropia | 3 (50%) | — |  |
| **Duration of thyroid disease prior to surgery (approx.mo)** | N/A | $112.2 \pm 195.97$ |  |
| **Duration of TED prior to surgery (approx.mo)** | N/A | $5.8 \pm 3.12$ |  |
| **Previous treatment for Grave's disease** |  |  |  |
| Antithyroid drugs | N/A | 4 (80%) |  |
| Thyroid surgery | N/A | 1 (20%) |  |
| Radioactive iodine therapy | N/A | 1 (20%) |  |
| **Previous treatment for TED** |  |  |  |
| Corticosteroid pulse therapy | N/A | 3 (60%) |  |
| Disarticulation of rectus | N/A | 2 (40%) |  |
| **Smoking history (n, %)** | 0 (0%) | 0 (0%) |  |
| **Exophthalmometry, hertel (mm)** | N/A | $19.9 \pm 4.72$ |  |
| **Presence of compressive optic neuropathy (n, %)** | N/A | 2 (40%) |  |
| **Surgery** |  |  |  |
| Orbital decompression | — | 5 (100%) |  |
| Blepharoplasty | 3 (50%) | — |  |
| Strabismus surgery | 3 (50%) | — |  |
| **Clinical activity score (0–7)** | N/A | $1.6 \pm 0.8$ |  |

**Notes.**
Abbreviations: N/A, not applicable; TED, Thyroid Eye Disease.
Data are shown as the mean ± SD.

SE, were analyzed. A significance threshold of *P* value < 0.01 was used to define differential alternative splicing events.

## Functional enrichment analysis

GO enrichment analyses for both the upregulated and downregulated genes were carried out using the R package topGO (*The Gene Ontology Consortium, 2021*) and the results were visualized using the REVIGO tool (http://revigo.irb.hr) (*Supek et al., 2011*). KEGG Orthology Based Annotation System (KOBAS) v3.0 (*Bu et al., 2021*) was used to perform the functional enrichment analysis. GSEA was carried out using the R package 'clusterProfiler' (*Yu et al., 2012*). The results are indicated in the appropriate figure legend and text.

## The protein–protein interaction (PPI) network and hub gene identification

Construction of a PPI network was conducted using STRING (https://string-db.org/). We uploaded DEGs to STRING and obtained high-resolution bitmaps. By calculating the

degree of connectivity, the hub genes in the PPI network were identified *via* cytoHubba, which is a plugin in Cytoscape software (version v3.9.1) (*Shannon et al., 2003*).

## RNA quantification

Total RNA was extracted using the RNAiso Plus (Takara, Kyoto, Japan), according to the manufacturer's instructions. Final RNA pellets were resuspended in nuclease-free H2O and then determine the purity and concentration by measuring the optical density at 260 nm and 280 nm (NanoDrop 2000c; Thermo Fisher Scientific, Waltham, MA, USA). Reverse transcription of total isolated RNA was performed using the PrimeScript RT master mix kit (Takara, Kyoto, Japan). Gene expression was measured by qRT-PCR. The data were analysed using the $2^{-\Delta\Delta CT}$ method and normalized to the endogenous control GAPDH mRNA (for humans), and the amount of target gene mRNA expression in each sample was expressed relative to that of the control. Primer sequences for qRT-PCR were designed using Primer Express Software (Thermo Fisher Scientific, Waltham, MA, USA; Table S2).

## Histological and immunohistochemical analysis

Human orbital adipose/connective tissues were obtained during orbital decompression and fixed overnight in 4% PFA (w/v) at 4 °C. The adipose sample was dehydrated through graded ethanol, and paraffin embedded. Histological sections of 5 μm were taken along the vertical meridian. Specimens were stained with H&E staining and observed under an Olympus BX-51 light microscope (Olympus, Tokyo, Japan). Standard immunohistochemical analysis with citrate antigen retrieval was performed with the antibodies against CD45 (#70257S; Cell Signaling, Danvers, MA, USA), Fibronectin (FN, #15613-1-AP; Proteintech, Chicago, IL, USA), and intercellular adhesion molecule 1 (ICAM1, #ab282575; Abcam, Cambridge, UK) to localize expression. Standard immunofluorescence analysis was performed to indicate F4/80 (#ab6640; Abcam, Cambridge, UK) expression, followed by Goat anti-Rabbit IgG (H+L) Cross-Adsorbed Secondary Antibody, Alexa Fluor™ 488 (#A-11008; Thermo Fisher Scientific, Waltham, MA, USA), and Goat anti-rat IgG (H+L) Cross-Adsorbed Secondary Antibody, Alexa Fluor™ 488 (#A-11006; Thermo Fisher Scientific, Waltham, MA, USA).

## Statistical analysis

The results are expressed as the mean ± SD. Significance was established between the two groups using Student's t test (paired *t* test). Age was compared using the *t*-test, and gender was compared using chi-squared tests. The data were analysed using GraphPad Prism 5 statistical software (Prism v5.0; GraphPad Software, La Jolla, CA, USA). A *P* value < 0.05 was considered statistically significant.

## RESULTS

### DEGs in orbital adipose/connective tissue samples of TAO patients

Deep sequencing identified 183 DEGs with the conditions of |log2(FC)| > 1 and *P* value < 0.05 between the orbital adipose/connective tissues of TAO patients and control individuals. Among these, 114 genes were upregulated, and 69 genes were downregulated. The fragments per kilobase million (FPKM) value of mRNAs shows that there is no

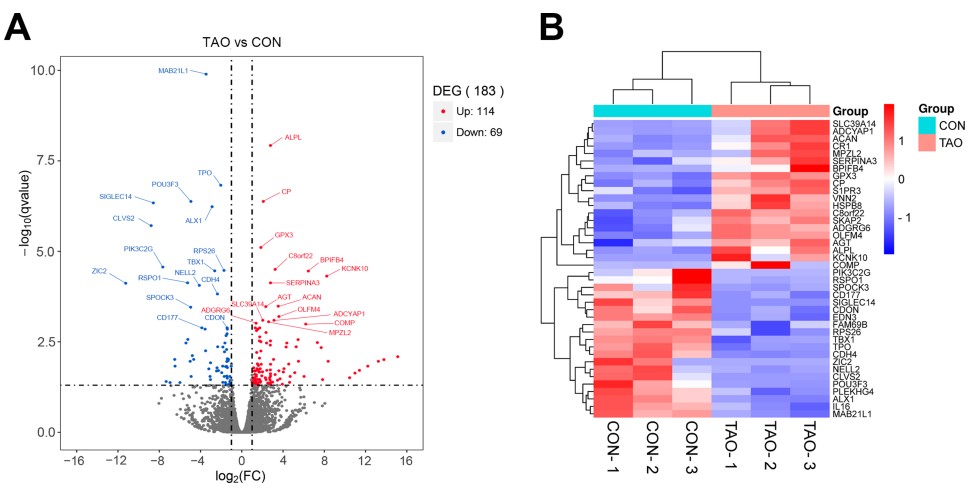

**Figure 1** **The differentially expressed genes were analyzed from RNA sequencing data.** (A) Volcano plot of different genes in control or TAO orbital fat. FC, fold change; DEGs, differentially expressed genes. (B) Hierarchical clustering heatmap showing gene expression differences.

abnormal expression in the three samples in each group (Figs. S2A–S2C). Principal component analysis (PCA) showed a significant separation between the two sets of samples (Fig. S2D). In our volcano plot and heatmap analysis of TAO-enriched genes, we showed the top 40 most DEGs in TAO samples compared to the controls (Figs. 1A, 1B). To identify and analyze the corresponding changes in these underlying functional DEGs, the enrichment analyses were employed.

## DAS gene analysis

Alternative splicing (AS) refers to the process of selectively removing or retaining exons/introns during the initial transcription of DNA into RNA and further processing into mature mRNA, resulting in multiple transcripts of a gene. To learn the potential AS of TAO patients, five main types of AS events were analyzed using rMATS, including exon skipping (SE), intron retention (RI), alternative 5′splice site (A5SS), alternative 3′splice site (A3SS), and mutually exclusive exons (MXE) (Fig. 2A). We selected the DAS genes with a threshold of $P$ value < 0.01. The numbers of A3SS, A5SS, MXE, RI, and SE events were 65, 57, 22, 18, and 477, respectively. SE was the most prevalent AS event in TAO patients, whereas RI was the least prevalent (Figs. 2B, 2C). This data suggests that an abnormal splicing process leads to specific splicing isoforms, which may have a close relationship with the occurrence and development of TAO.

## Enrichment analyses of DEGs

To explore the functions of DEGs, functional enrichment analysis was performed on DEGs by linking them with biological phenomena and their underlying mechanisms.

### GO annotation analyses

GO analysis is a common useful method for large-scale functional enrichment research, which can significantly distribute DEGs into the biological process (BP), molecular function

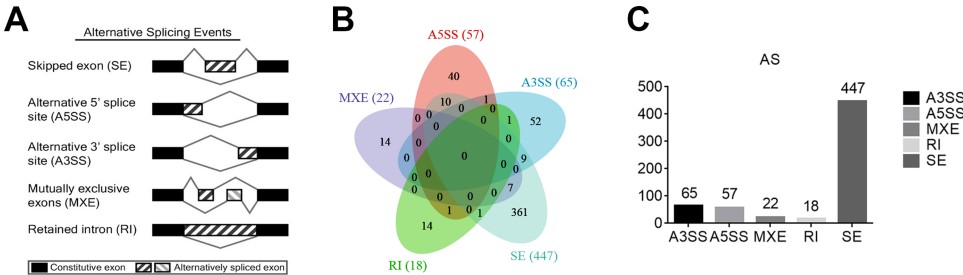

**Figure 2  Analysis of differential alternative splicing (AS) genes and distribution of the five main AS events.** (A) Schematic diagrams of the mechanisms of the five main AS events. (B) Venn diagram of the detected genes undergoing the five AS events and overlap of these genes. SE, exon skipping; RI, intron retention; A5SS, alternative 5′ splice site; A3SS, alternative 3′ splice site; MXE, mutually exclusive exons. (C) Distribution of differential AS events based on a threshold of $P < 0.01$.

(MF), and the cellular component (CC). The most significant GO terms of upregulated and downregulated DEGs are shown in Figs. 3A–3C, and detailed information is listed in Table 2.

In the GO terms of TAO samples, inflammation response was the main BP category, including inflammatory response, regulation of inflammatory response, acute inflammatory response, regulation of acute inflammatory response, and myeloid leukocyte migration (Fig. 3A). This suggests that the pathogenesis of TAO is closely related to the aberrant activation of inflammatory responses, which play a key role in the activation of orbital adipogenesis. The MF category was abundant in glycosaminoglycan binding, G protein-coupled receptor binding, signaling receptor binding, and extracellular matrix structural constituent (Fig. 3B). In addition, CC mainly displayed extracellular region, extracellular space, and cell surface (Fig. 3C).

### KEGG pathway enrichment analyses

The KEGG database is a widely used database to systematically analyze the metabolic pathways of gene products in cells and the functions of these gene products. It can help us study genes and expression information as a whole network. By analyzing the signaling pathway of DEGs, we can understand the significantly changed metabolic pathway in the state of TAO, which is important for exploring the pathogenesis of the disease.

KEGG analysis showed that 142 pathways were significantly enriched. The top 20 enriched pathways are shown in Fig. 3D. The represented pathways were ECM-receptor interaction, PI3K-Akt signaling pathway, cell adhesion molecules, cytokine–cytokine receptor interaction, and focal adhesion.

### GSEA

GSEA is a promising, widely used software package that derives gene sets to determine different biological functions between two groups. By GSEA, we identified that cytokine–cytokine receptor interaction, cytokine–cytokine receptor interaction, NF-kappa B signaling pathway, rheumatoid arthritis, TNF signaling pathway, and viral protein interaction with cytokine and cytokine receptor were the top five enriched pathways

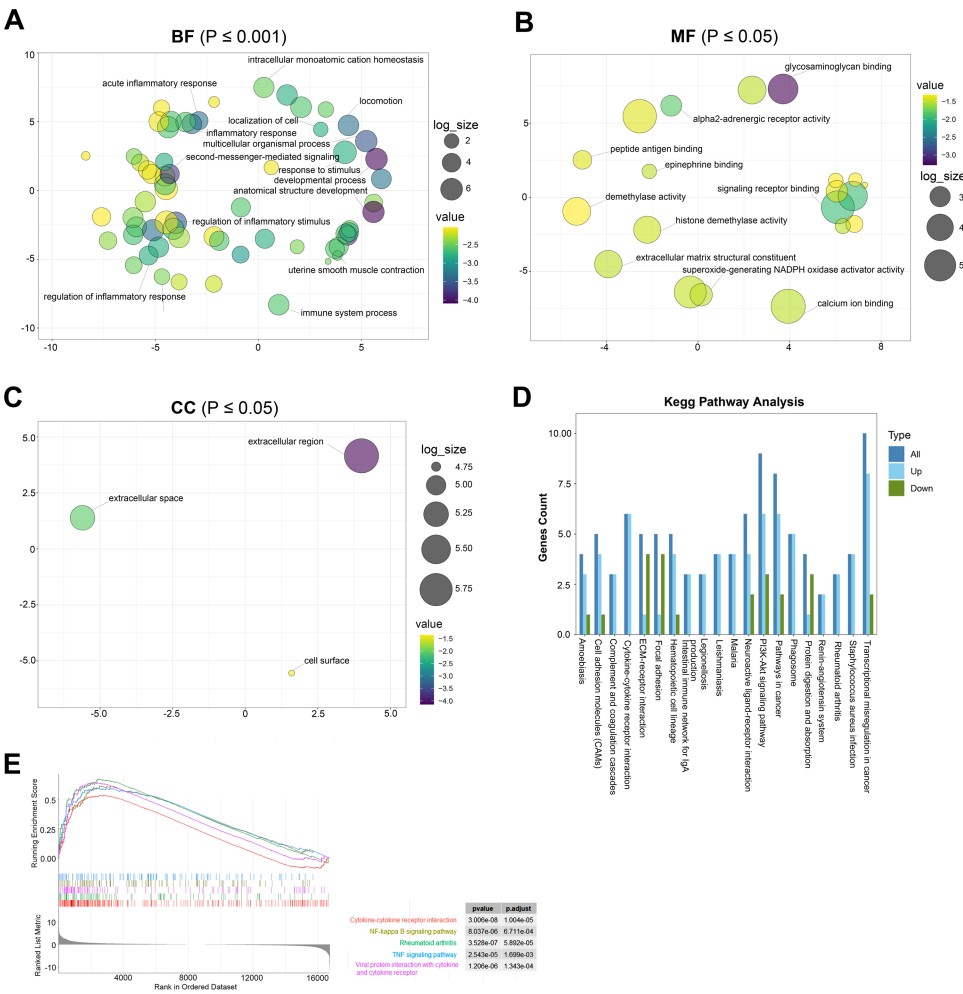

**Figure 3  The most significantly enriched GO terms and KEGG pathway analysis relevant to up- and downregulated genes.** (A) BP term of GO enrichment analysis, *$p < 0.001$. BP, biological process. (B) MF term of GO enrichment analysis, *$p < 0.05$. MF: the molecular function. (C) CC term of GO enrichment analysis, *$p < 0.05$. CC: cellular component. (D) KEGG pathway analysis showing pathways that are enriched in the TAO group. (E) Gene cluster enrichment analysis (GSEA) revealed a significant enrichment of the first five pathways in TAO patients.

(Fig. 3E). In summary, the biological processes from the enriched GO terms, KEGG pathways, and GSEA for the DEGs were mainly involved in the regulation of inflammatory response, glycosaminoglycan binding and hyaluronic acid binding.

## Cross with gene expression omnibus (GEO) database
We downloaded the microarray data of GSE185952 from the GEO database (*Yue et al., 2021*). This dataset contains six samples, including three TAO patients who underwent orbital decompression for proptosis correction and three control groups obtained from patients who underwent plastic surgery. We screened out the DEGs on the cut-off criteria with |log2 (FC)| > 1 and *P* value < 0.01. Intersection analysis was performed on the DEGs of the two independent samples. We obtained six co-upregulated genes, cartilage oligomeric

**Table 2  The top GO terms of DEGs between TAO and control samples.** The top 10 BP terms, MF terms and the most significantly CC terms of DEGs between TAO and control samples.

| Category | ID | Term | Gene |
|---|---|---|---|
| BP | GO:0007275 | multicellular organism development | ACAN\|ADAMTS18\|ADAMTS9\|ADCYAP1\|ADGRG6\| ADRA2B\|AGT\|ALPL\|ALX1\|AQP3\|ARID5B\|BAIAP2\|BMP3\| C8orf22\|CDH11\|CDH4\|CDON\|COL9A3\|COMP\|CP\|CXCL8\| CXCR4\| CYP19A1\|DUSP2\|EDN3\|EDNRB\|EFEMP1\|EGFL6\|FAP\| FGF1\| FOXD1\|FOXN4\|GABRA4\|GATA6\|GFRA1\|HIF1A\|HIF3A \|HMOX1\| HOXC9\|ICOS\|IHH\|KCNA1\|KRT25\|LCP1\|LFNG\| MAB21L1\| MCL1\|MCOLN3\|MEIS1\|NGFR\|NLGN4Y\|NOCT\| NR2F1\|NTS\| PAPPA2\|PCSK6\|PDE4D\|PFKFB3\|PHLDA1\| PKP2\|PLEK\| POU3F3\|PPL\|PTHLH\|RPS4Y1\|RUNX1\| S100A9\|S1PR3\|SFRP4\|SHC3\|SLC7A5\|SPRY4\|T\| TBX1\|TBX3\|TENM1\|TMEM176A\|TPO\|TRIB1\| USP9Y\|VCAN\|VCX\|VNN2\|WNT5B\| XIRP1\|ZFY\|ZIC1\|ZIC2 |
| BP | GO:0048856 | anatomical structure development | ACAN\|ADAMTS18\|ADAMTS9\|ADCYAP1\|ADGRG6\| ADRA2B\|AGT\|ALPL\|ALX1\|AQP3\|ARID5B\|BAIAP2\| BMP3\|C8orf22\|CDH11\|CDH4\|CDON\|COCH\|COL9A3\| COMP\|CP\|CXCL8\|CXCR4\|CYP19A1\|DNASE1L3\|DUSP2\| EDN3\|EDNRB\|EFEMP1\|EGFL6\|FAP\|FGF1\|FOXD1\|FOXN4\| GABRA4\|GATA6\|GFRA1\|HIF1A\|HIF3A\|HMOX1\|HOXC9\| ICOS\|IHH\|KCNA1\|KRT25\|LCP1\|LFNG\|MAB21L1\|MCL1\| MCOLN3\|MEIS1\|MPZL2\|NGFR\|NLGN4Y\|NOCT\|NR2F1\| NTS\|OLFM4\|PAPPA2\|PCSK6\|PDE4D\|PFKFB3\|PHLDA1\| PKP2\|PLEK\|POU3F3\|PPL\|PTHLH\|RPS4Y1\|RUNX1\|S100A9\| S1PR3\|SFRP4\|SHC3\|SLC7A5\|SPRY4\|T\|TBX1\|TBX3\| TENM1\|TMEM176A\|TPO\|TRIB1\|UGCG\|USP9Y\|VCAN\|VCX\| VNN2\|WNT5B\|XIRP1\|ZFY\|ZIC1\|ZIC2 |
| BP | GO:0032502 | developmental process | ACAN\|ADAMTS18\|ADAMTS9\|ADCYAP1\|ADGRG6\| ADRA2B\|AGT\|ALPL\|ALX1\|AQP3\|ARID5B\|BAIAP2\| BMP3\|C8orf22\|CDH11\|CDH4\|CDON\|COCH\|COL9A3\| COMP\|CP\|CXCL8\|CXCR4\|CYP19A1\|DDX21\|DNASE1L3\| DUSP2\|EDN3\|EDNRB\|EFEMP1\|EGFL6\|FAP\|FGF1\|FNDC5\| FOXD1\|FOXN4\|GABRA4\|GATA6\|GFRA1\|HIF1A\|HIF3A\| HMOX1\|HOXC9\|ICOS\|IHH\|KCNA1\|KRT25\|LCP1\|LFNG\| MAB21L1\|MCL1\|MCOLN3\|MEIS1\|MPZL2\|MSR1\|NEK6\| NGFR\|NLGN4Y\|NOCT\|NR2F1\|NTS\|OLFM4\|PAPPA2\|PCSK6\| PDE4D\|PFKFB3\|PHLDA1\|PKP2\|PLEK\|POU3F3\|PPL\| PTHLH\|RPS4Y1\|RUNX1\|S100A9\|S1PR3\|SFRP4\|SHC3\| SLC7A5\|SPRY4\|T\|TBX1\|TBX3\|TENM1\|TMEM176A\|TPO\| TRIB1\|UGCG\|USP9Y\|VCAN\|VCX\|VNN2\|WNT5B\| XIRP1\|ZFY\|ZIC1\|ZIC2 |
| BP | GO:0001501 | skeletal system development | ACAN\|ALPL\|ALX1\|ARID5B\|BMP3\|CDH11\| COMP\|EFEMP1\|HIF1A\|HOXC9\|IHH\|MEIS1\| PAPPA2\|PTHLH\|RUNX1\|T\|TBX1\| TBX3\|VCAN\|WNT5B |
| BP | GO:0019932 | second-messenger-mediated signaling | ADCYAP1\|ADGRG6\|ADRA2A\|ADRA2B\|AGT\| CCL4\|CXCL8\|CXCR4\|EDN3\|EDNRB\|FPR1\| GPR3\|PDE4D\|PLEK\|PTHLH |

**Table 2** (*continued*)

| Category | ID | Term | Gene |
|---|---|---|---|
| BP | GO:0032501 | multicellular organismal process | ACAN\|ACE2\|ADAMTS18\|ADAMTS9\|ADCYAP1\|ADGRG6\|ADRA2A\|ADRA2B\|AGT\|ALPL\|ALX1\|AQP3\|ARID5B\|BAIAP2\|BMP3\|C8orf22\|CD177\|CDH11\|CDH4\|CDON\|COCH\|COL9A3\|COMP\|CP\|CXCL8\|CXCR4\|CYP19A1\|DDX21\|DDX3Y\|DUSP2\|EDN3\|EDNRB\|EFEMP1\|EGFL6\|F13A1\|FAM107B\|FAP\|FGF1\|FOSB\|FOXD1\|FOXN4\|GABRA4\|GATA6\|GFRA1\|GP1BB\|HIF1A\|HIF3A\|HILPDA\|HMOX1\|HOXC9\|ICOS\|IHH\|KCNA1\|KCNK10\|KRT25\|LCP1\|LFNG\|MAB21L1\|MCL1\|MCOLN3\|MEIS1\|MLIP\|MMRN1\|MSR1\|MYH2\|NGFR\|NLGN4Y\|NOCT\|NR2F1\|NTS\|PAPPA2\|PCSK6\|PDE4D\|PFKFB3\|PHLDA1\|PKP2\|PLEK\|POU3F3\|PPL\|PRKAR2B\|PTHLH\|RPS4Y1\|RUNX1\|S100A9\|S1PR3\|SAA1\|SERPINA3\|SFRP4\|SHC3\|SLC7A5\|SPRY4\|T\|TBX1\|TBX3\|TENM1\|TLR2\|TMEM176A\|TPO\|TRIB1\|USP9Y\|VCAN\|VCX\|VNN2\|WNT5B\|XIRP1\|ZFY\|ZIC1\|ZIC2 |
| BP | GO:0048518 | positive regulation of biological process | ACE2\|ADAMTS9\|ADCYAP1\|ADRA2A\|ADRA2B\|AGT\|ALX1\|AQP3\|ARID5B\|BAIAP2\|BCL2A1\|BMF\|BMP3\|C4A\|CCL4\|CDH4\|CDON\|COCH\|CR1\|CXCL8\|CXCR4\|DCUN1D3\|DDX21\|DDX3Y\|EDN3\|EDNRB\|EFEMP1\|EGFL6\|FAP\|FCGR1A\|FGF1\|FNDC5\|FOSB\|FOXD1\|FOXN4\|FPR1\|GATA6\|HIF1A\|HIF3A\|HILPDA\|HLA-DRB5\|HMOX1\|ICOS\|IHH\|IL16\|LCP1\|LFNG\|MAB21L1\|MCL1\|MEIS1\|MSR1\|MYH2\|NEK6\|NGFR\|NOCT\|NR2F1\|OLFM4\|OSMR\|PDE4D\|PHLDA1\|PKP2\|PLEK\|POU3F3\|PRKAR2B\|PTHLH\|RNASE2\|RPS4Y1\|RSPO1\|RUNX1\|S100A9\|S1PR3\|SAA1\|SAMD4A\|SEPT5\|SFRP4\|SKAP2\|SLA\|SLC30A8\|T\|TBX1\|TBX3\|TENM1\|TLR2\|TRIB1\|UGCG\|WNT5B\|ZIC1\|ZIC2 |
| BP | GO:0048731 | system development | ACAN\|ADAMTS18\|ADCYAP1\|ADGRG6\|ADRA2B\|AGT\|ALPL\|ALX1\|AQP3\|ARID5B\|BAIAP2\|BMP3\|C8orf22\|CDH11\|CDH4\|CDON\|COL9A3\|COMP\|CP\|CXCL8\|CXCR4\|CYP19A1\|EDN3\|EDNRB\|EFEMP1\|FAP\|FGF1\|FOXD1\|FOXN4\|GABRA4\|GATA6\|GFRA1\|HIF1A\|HIF3A\|HMOX1\|HOXC9\|ICOS\|IHH\|KCNA1\|KRT25\|LCP1\|LFNG\|MAB21L1\|MCOLN3\|MEIS1\|NGFR\|NLGN4Y\|NR2F1\|NTS\|PAPPA2\|PDE4D\|PFKFB3\|PHLDA1\|PKP2\|PLEK\|POU3F3\|PPL\|PTHLH\|RUNX1\|S100A9\|SHC3\|SLC7A5\|T\|TBX1\|TBX3\|TENM1\|TMEM176A\|TPO\|TRIB1\|USP9Y\|VCAN\|VCX\|VNN2\|WNT5B\|XIRP1\|ZIC1\|ZIC2 |
| BP | GO:0040011 | locomotion | ADRA2A\|AGT\|ALX1\|ARID5B\|CCL4\|CD177\|CDH4\|CXCL8\|CXCR4\|CYP19A1\|EDN3\|EDNRB\|EFEMP1\|FAP\|FGF1\|FOXD1\|FPR1\|HIF1A\|HMOX1\|IL16\|LCP1\|NGFR\|NR2F1\|OLFM4\|PDE4D\|PIK3C2G\|PKP2\|POU3F3\|RNASE2\|S100A9\|SAA1\|SLC7A5\|T\|TBX1\|TRIB1\|USP9Y\|VCAN\|WNT5B |
| BP | GO:0006954 | inflammatory response | ACE2\|ADCYAP1\|ADRA2A\|AGT\|C4A\|CCL4\|CR1\|CXCL8\|CXCR4\|CYP19A1\|EDNRB\|FCGR1A\|HIF1A\|HMOX1\|NFKBIZ\|NGFR\|OSMR\|S100A9\|S1PR3\|SAA1\|SERPINA3\|TLR2 |

**Table 2** (*continued*)

| Category | ID | Term | Gene |
|---|---|---|---|
| MF | GO:0005539 | glycosaminoglycan binding | ACAN\|COMP\|FGF1\|NELL2\|PCSK6\|RSPO1\|SAA1\|SPOCK3\|SUSD5\|TENM1\|TLR2\|VCAN |
| MF | GO:0001664 | G protein-coupled receptor binding | ADCYAP1\|ADRA2A\|AGT\|CCL4\|CXCL8\|EDN3\|EDNRB\|PDE4D\|RSPO1\|SAA1\|WNT5B |
| MF | GO:0005102 | signaling receptor binding | ADCYAP1\|ADRA2A\|AGT\|BMP3\|CCL4\|CXCL8\|EDN3\|EDNRB\|EFEMP1\|EGFL6\|FAP\|FGF1\|FNDC5\|FPR1\|GFRA1\|HIF1A\|HILPDA\|IHH\|IL16\|MICA\|NGFR\|NLGN4Y\|NTS\|PDE4D\|PTHLH\|RSPO1\|S100A9\|S1PR3\|SAA1\|SHC3\|TLR2\|WNT5B |
| MF | GO:0004938 | alpha2-adrenergic receptor activity | ADRA2A\|ADRA2B |
| MF | GO:0016176 | superoxide-generating NADPH oxidase activator activity | AGT\|NOXA1 |
| MF | GO:0051379 | epinephrine binding | ADRA2A\|ADRA2B |
| MF | GO:0005201 | extracellular matrix structural constituent | ACAN\|COL4A6\|COL9A3\|COMP\|VCAN |
| MF | GO:0008201 | heparin binding | COMP\|FGF1\|NELL2\|PCSK6\|RSPO1\|SAA1\|TENM1 |
| MF | GO:0005540 | hyaluronic acid binding | ACAN\|SUSD5\|VCAN |
| MF | GO:0048018 | receptor ligand activity | ADCYAP1\|AGT\|BMP3\|CCL4\|CXCL8\|EDN3\|EFEMP1\|FGF1\|FNDC5\|IL16\|NTS\|PTHLH\|SAA1 |
| CC | GO:0005576 | extracellular region | ACAN\|ACE2\|ADAMTS9\|ADCYAP1\|AGT\|ALPL\|BAIAP2\|BMP3\|BPIFB4\|C4A\|CCL4\|CD177\|CDH11\|CDON\|COCH\|COL4A6\|COMP\|CP\|CPXM1\|CR1\|CXCL8\|CXCR4\|DDX3Y\|EDN3\|EFEMP1\|EGFL6\|F13A1\|FAP\|FGF1\|FNDC5\|GFRA1\|GPX3\|HILPDA\|HLA-DRB5\|HMCN2\|HMOX1\|ICOS\|IGLON5\|IHH\|IL16\|IL1R2\|KRT25\|KSR2\|LCP1\|LFNG\|MICA\|MLPH\|MMRN1\|MSR1\|NELL2\|NGFR\|NLGN4Y\|NTS\|OLFM4\|PAPPA2\|PCSK1\|PCSK6\|PI3\|PLEK\|PPL\|PRKAR2B\|PTHLH\|RNASE2\|RPS26\|RPS4Y1\|RSPO1\|S100A9\|SAA1\|SCG5\|SERPINA3\|SFRP4\|SLC7A5\|SPOCK3\|TENM1\|TPO\|VCAN\|WNT5B |
| CC | GO:0044421 | extracellular region part | ACAN\|ACE2\|ADAMTS9\|ADCYAP1\|AGT\|ALPL\|BAIAP2\|BMP3\|C4A\|CCL4\|CD177\|CDH11\|CDON\|COCH\|COL4A6\|COMP\|CP\|CPXM1\|CR1\|CXCL8\|CXCR4\|DDX3Y\|EDN3\|EFEMP1\|EGFL6\|F13A1\|FAP\|FGF1\|GFRA1\|GPX3\|HILPDA\|HLA-DRB5\|HMCN2\|HMOX1\|IHH\|IL16\|KRT25\|KSR2\|LCP1\|LFNG\|MICA\|MLPH\|MSR1\|NELL2\|NLGN4Y\|OLFM4\|PAPPA2\|PCSK1\|PCSK6\|PI3\|PPL\|PRKAR2B\|PTHLH\|RNASE2\|RPS26\|RPS4Y1\|S100A9\|SAA1\|SERPINA3\|SFRP4\|SLC7A5\|SPOCK3\|TPO\|VCAN\|WNT5B |

**Table 2** (*continued*)

| Category | ID | Term | Gene |
|---|---|---|---|
| CC | GO:0005615 | extracellular space | ACE2\|ADAMTS9\|ADCYAP1\|AGT\|ALPL\|BAIAP2\|BMP3\|C4A\|CCL4\|CD177\|CDH11\|COCH\|COMP\|CP\|CPXM1\|CR1\|CXCL8\|CXCR4\|DDX3Y\|EDN3\|EFEMP1\|EGFL6\|F13A1\|FAP\|FGF1\|GFRA1\|GPX3\|HILPDA\|HLA-DRB5\|HMOX1\|IHH\|IL16\|KRT25\|KSR2\|LCP1\|MICA\|MLPH\|MSR1\|NELL2\|NLGN4Y\|OLFM4\|PAPPA2\|PCSK1\|PCSK6\|PI3\|PPL\|PRKAR2B\|PTHLH\|RNASE2\|RPS26\|RPS4Y1\|S100A9\|SAA1\|SERPINA3\|SFRP4\|SLC7A5\|SPOCK3\|TPO\|VCAN\|WNT5B |
| CC | GO:0009986 | cell surface | ACE2\|ADAMTS9\|CDON\|CR1\|CXCR4\|FAP\|FCGR1A\|HILPDA\|ICOS\|KCNA1\|MICA\|NGFR\|NLGN4Y\|PCSK6\|SFRP4\|TLR2\|TPO\|WNT5B |

**Notes.**

Abbreviations: BP, biological process; MF, molecular function; CC, cellular component.
Cut-off criteria: $*p < 0.05$.

matrix protein(COMP), interleukin-8(CXCL8), fmet-leu-phe receptor (FPR1), chemokine (c-c motif) ligand 4-like 1 (CCL4), protein s100-a9 (S100A9), and nf-kappa-b inhibitor zeta isoform x2 (NFKBIZ), and 14 co-downregulated genes, alx homeobox protein 1 (ALX1), protein kinase c-binding protein nell2 isoform x3 (NELL2), cadherin-4 isoform x1 (CDH4), puratrophin-1 isoform x1 (PLEKHG4), cochlin (COCH), low-quality protein: bcl-2-modifying factor (BMF), probable carboxypeptidase x1 (CPXM1), prolyl endopeptidase fap isoform x1 (FAP), low quality protein: hemicentin-2 (HMCN2), melanophilin isoform x1(MLPH), collagen alpha-6 chain (COL6A6), epidermal growth factor-like protein 6 isoform x1 (EGFL6), shc-transforming protein 3 (SHC3), and periplakin (PPL) (Fig. 4A).

## The protein–protein interaction (PPI) network and hub gene

To better understand the molecular mechanism of TAO, we visualized the importance of the relationship between proteins of DEGs using Cytoscape software (Table S3, Fig. 4B). Moreover, we identified the top 10 HUB genes according to node degree among these target genes *via* the Cytoscape plug-in cytoHubba (Fig. 4C). Collectively, these results suggest that the core proteins CXCL8, Toll-like receptor-2 (TLR2), CCL4 and angiotensinogen (AGT) in the PPI network may be involved in the regulation of TAO pathogenesis.

## Validation of the expression of DEGs

We confirmed the DEGs by qRT-PCR in orbital adipose/connective tissues to confirm the results of RNA-seq. In orbital adipose tissues, alkaline tissue-nonspecific isozyme isoform x1 (ALPL), ceruloplasmin isoform x3 (CP), and AGT were significantly upregulated, and protein mab-21-like 1 (MAB21L1), phosphoinositide 3-kinase gamma-subunit (PIK3C2G), and clavesin-2 (CLVS2) were significantly downregulated compared with controls (Figs. 5A, 5B). In orbital muscle, glutathione peroxidase 3 (GPX3) and alpha-1-antichymotrypsin isoform x1 (SERPINA3) were upregulated, while MAB21L1 and PIK3C2G were downregulated, but the difference did not achieve statistical significance (Figs. 5C, 5D). Among these genes, only R-spondin 1 (RSPO1) was significantly
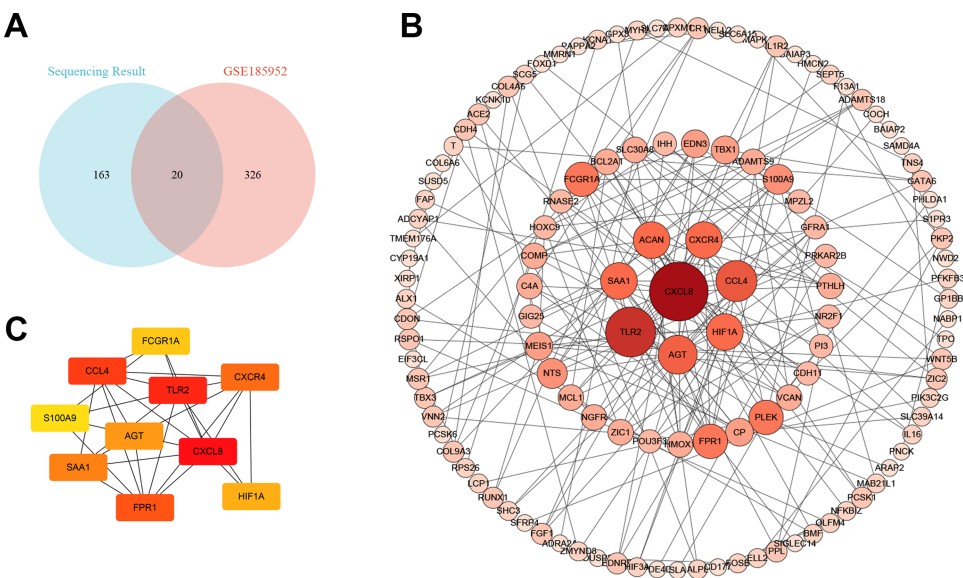

**Figure 4  The Venn diagram and the top hub genes identified in the protein–protein interaction (PPI) networks.** (A) The Venn diagram shows the differentially expressed gene identification in the two gene expression profile datasets. (B) PPI network of differentially expressed genes. (C) Identification of the top 10 hub genes.

downregulated in orbital muscle tissues. Thus, the validation results by qRT-PCR are consistent with the RNA sequencing results.

## Histology and inflammation in the orbital adipose/connective tissues of TAO patients and control individuals

H&E staining showed the morphology of orbital adipose tissue, and consistently indicated an increased level of the inflammatory cell infiltration (black arrows) in TAO patients compared with the control individuals (Fig. 6A). Meanwhile, we immunohistochemically stained sections for the CD45, a protein expressed on all leukocytes, and found that CD45 expression (black arrows) also increased in the TAO patients compared with controls (Fig. 6B). To identify and quantitate macrophages within adipose tissue, we detected the expression of F4/80 antigen, a marker specific for mature macrophages in the orbital adipose tissues. Indeed, the TAO group had significantly increased amounts of F4/80-positive macrophages, compared with the control groups.

Besides, in the orbital muscle tissues, the inflammatory markers, CD45 and ICAM1 also increased surrounding the myofibrils in patients with TAO compared with the control groups (Figs. S3A and S3B). Moreover, there was a potent increase in the expression of fibrotic proteins, including $\alpha$-SMA and FN (Figs. S3C and S3D), indicating the orbital fibrosis or myositis in individuals with TAO.

Collectively, our results showed that there were enhanced inflammatory responses in orbital adipose/connective tissue and increased levels of fibrosis in the extraocular muscles among TAO patients.

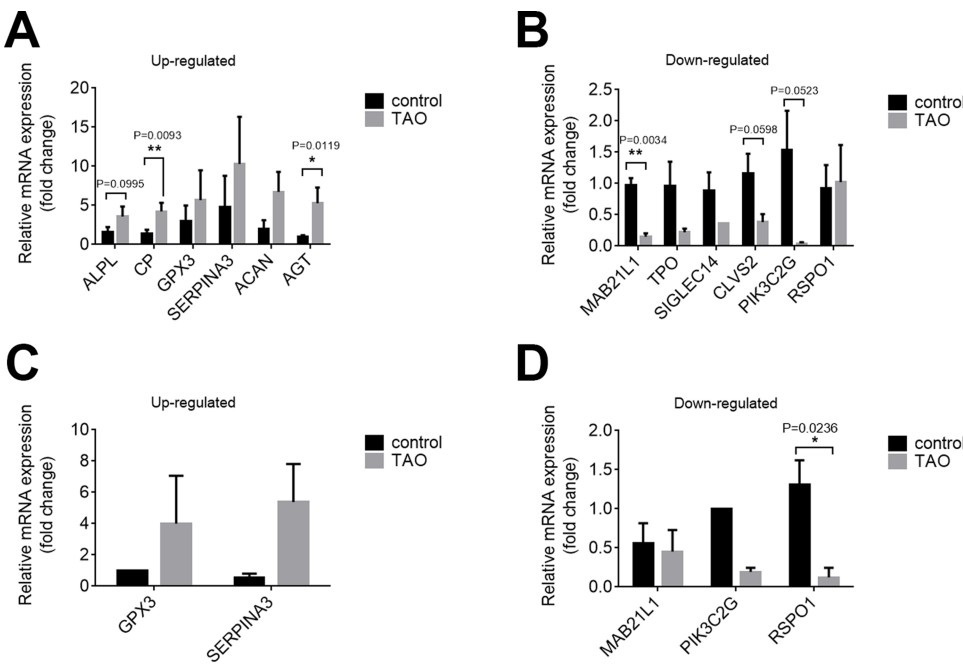

**Figure 5** **Validation of the expression levels of mRNAs in the TAO groups and control groups.** (A and B) The mRNA expression levels in orbital adipose tissue as verified by qRT–PCR. (C and D) Expression levels of mRNAs in orbital muscle tissues as verified by qRT–PCR. The results are presented as the means ± SDs; $n = 4$, $*$ $p < 0.05$, and $**p < 0.01$ for each pair of groups indicated.

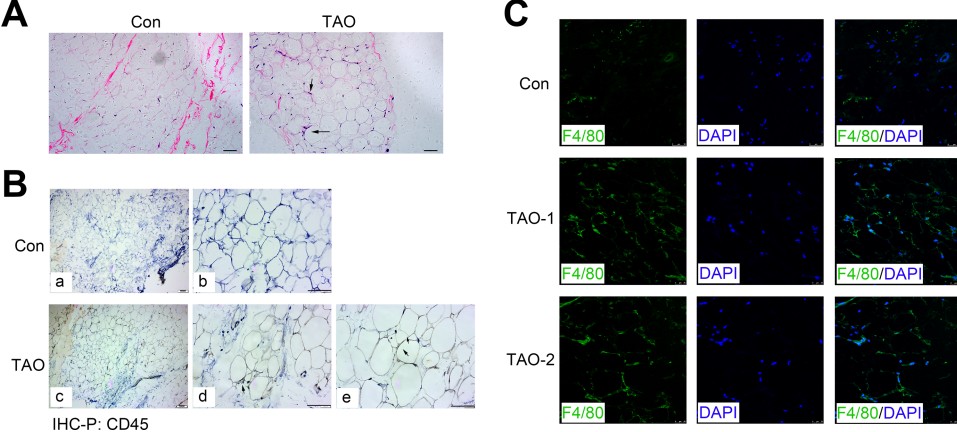

**Figure 6** **Orbital adipose tissue inflammation in the TAO patients and control individuals.** (A) H & E staining in paraffin sections of orbital fat; scale bar, 25 μm. (B) Immunohistochemistry for CD45 (black arrows) and a hematoxylin nuclear counterstain (blue) was performed on orbital adipose tissue; scale bar, 25 μm. (C) Immunofluorescence detection of the macrophage-specific antigen F4/80 (green) in orbital adipose tissue from TAO patients and control individuals; scale bar, 25 μm.

## DISCUSSION

TAO is an autoimmune disease that affects orbital adipose tissue and extraocular muscles (*Bartalena & Tanda, 2022*). To date, the pathogenic mechanisms of TAO have not been clearly understood. Symptomatic treatments, such as hormone pulse therapy and orbital decompression, are currently limited for patients with TAO (*Baeg et al., 2022*). This issue emphasizes the importance of understanding the underlying mechanism(s) of and identifying therapeutic approaches for the prevention or treatment of TAO. In this study, we analysed the DEGs in orbital adipose/connective tissue from TAO patients and controls. The symptoms of TAO are mainly caused by the inflammation in the orbital connective tissue, an increase in orbital volume due to enhanced adipogenesis and overproduction of glycosaminoglycans, and fibrosis of the extraocular muscles (*Kahaly et al., 1992*).

It has been reported that the inflammatory levels significantly upregulated in the adipose tissue and muscle of TAO patients (*Carroll et al., 2013*; *Natesha et al., 1992*; *Khong et al., 2015*). *Huang et al. (2022)* demonstrated that endoplasmic reticulum stress initiated by cholesterol metabolism may provoke adipose inflammation in TAO. Adipocyte-derived CP and AGT play a critical role in adipogenesis as well as inflammation (*Carroll et al., 2013*; *Bednarek, Wysocki & Sowinski, 2004*). Consistent with previous studies, we found elevated levels of CP and ATG in the adipose tissue and muscle of TAO patients. Existing data show that SERPINA3, an acute phase response protein, is involved in the pathogenesis of acute anterior uveitis, chronic obstructive pulmonary disease, Parkinson's disease, Alzheimer's disease, and coronary artery disease (*Eidet et al., 2021*; *Li et al., 2021*; *Sánchez-Navarro et al., 2021*). There is also literature supporting that SERPINA3 can be expressed to promote cell proliferation, migration, and expression of inflammatory cytokines by NF-κB signaling pathways (*Liu et al., 2022*). Consistently, SERPINA3 is also upregulated in both adipose tissue and muscle in TAO. Our research also combined RNA sequencing analysis with multiple validation experiments including qRT-PCR, H&E, immunohistochemistry and immunofluorescence analysis. H&E staining, CD45 and ICAM1 immunohistochemistry staining, and F4/80 immunofluorescence staining results showed the inflammatory responses potently increased in the orbital adipose/connective tissues of TAO patients, compared with the control groups (Figs. 6A and 6B, Figs. S3A and S3B).

In our study, RSPO1 was downregulated more significantly in orbital connective tissue than that in orbital fatty tissue. We speculated that this may be related to the fibrosis of the extraocular muscles. There is literature indicating that in other organs, such as the kidney, RSPO1 plays an important role in fibrogenesis, which may explain why the downward trend of RSPO1 is more pronounced in muscles (*Su et al., 2021*).

In our study, the results of GO molecular function analysis indicated that these DEGs were enriched in several terms, such as glycosaminoglycan binding, and extracellular matrix structural constituent. *Wu et al. (2020)*, *Wu et al. (2021a)* and *Wu et al. (2021b)* indicated that several extracellular matrix related mRNAs (such as COL12A1, COL6A3) significantly reduced in TAO samples and closely related to the abnormal deposition of the extracellular matrix in orbital fat tissues in TAO patients (*Liang et al., 2021*). Additionally, GSEA and KEGG pathway enrichment analyses of the DEGs also showed marked enrichment of

the NF-κB pathway, ECM-receptor interaction, cell adhesion molecules, and PI3K-Akt signaling pathway. During the pathogenesis of TAO, orbital fibroblasts are thought to interact with immunocompetent cells recruited to the orbit (*Heufelder, 1995*). They produce high amounts of glycosaminoglycans, particularly hyaluronan, which absorb water and lead to an increase in matrix volume (*Smith et al., 1991*). It has been documented previously that CD40–CD40 ligand interactions have important roles in the activation of human orbital fibroblasts (*Cao et al., 1998*). CD40L-provoked signaling pathways, including the NF-kappa B pathway and PI3K-Akt signaling pathway, result in the high expression of a variety of cytokines, such as VCAM-1, E-selective protein, IL-6, and other cytokines, in orbital fibroblasts of patients with TAO (*Gillespie et al., 2012*; *Hwang et al., 2009*). Fibroblasts are reported to be responsible for the secretion of collagen, release of extracellular matrix, and participation in inflammatory responses (*Smith & Janssen, 2019*). This functional characterization is further substantiated by $\alpha$-SMA and FN immunofluorescent staining results (Figs. S3C and S3D).

As a previous study reported, there may exist alterations in the composition of the intestinal microbiota among patients, who suffered from severe and active TAO (*Mori, Nakagawa & Ozaki, 2012*). We found that pathway analyses highlighted the enrichment of highly expressed genes in the intestinal immune network for IgA production. In a separate investigation, *Shi et al. (2019)* found that two gut microbiotas (s_Prevotella_copri and f_Prevotellaceae) showed a significant correlation with TRAb. This suggests that intestinal symbiotic microorganisms may influence extraintestinal immune responses through the mucosal immune response induced by IgA antibodies, and they may render tolerance to self-antigens incompetent, such as TRAb, which can stimulate orbital and periorbital tissues and constitutes an independent risk factor for GO (*Pianta et al., 2017*; *Seo & Sanchez Robledo 2018*).

As with all transcriptomic analyses, there are limitations to this study. With the use of human tissue, there is heterogeneity in the patient's genetic background and other characteristics, such as age, gender, and CAS, which likely affect the disease. As such, we removed the influence of smoking on our results as much as possible, which has a strong and consistent association with TAO (*Bartalena et al., 1989*). One notable limitation lies in the relatively small sample size employed in our study, which consequently limits the statistical power. Additionally, while we selected genes that we believed were most important to the pathogenic mechanisms of TAO, it is imperative to acknowledge the presence of numerous other DEGs and pathways presented in these results that could be important and contribute to TAO.

## CONCLUSIONS

Our transcriptome analysis identified 183 DEGs between TAOs and normal orbit tissues. Through an integrated bioinformatics analysis and verification of the DEGs, we identified several key candidate genes and enriched pathways that may aid the search for biomarkers

and therapeutic targets of TAO. However, further molecular biology experiments are required to validate the findings of this study.

### Funding

This project was supported by the National Natural Science Foundation of China (81800845, 81770941), the Wuxi Taihu Lake Talent Plan, Supports for Leading Talents in Medical and Health Profession (grant Numbers: 2020-THRCTD-1, THRC-DJ-1), the Medial Key Discipline Program of Wuxi Health Commission (ZDXK2021001) and the Top Talent Support Program for young and middle-aged people of Wuxi Health Committee (grant Numbers: HB2020004, HB2020022).There was no additional external funding received for this study. The funders had no role in study design, data collection and analysis, decision to publish, or preparation of the manuscript.

### Grant Disclosures

The following grant information was disclosed by the authors:
National Natural Science Foundation of China: 81800845, 81770941.
Wuxi Taihu Lake Talent Plan, Supports for Leading Talents in Medical and Health Profession: 2020-THRCTD-1, THRC-DJ-1.
Medial Key Discipline Program of Wuxi Health Commission: ZDXK2021001.
Top Talent Support Program for young and middle-aged people of Wuxi Health Committee: HB2020004, HB2020022.

### Competing Interests

The authors declare there are no competing interests.

### Author Contributions

- Yan Wang conceived and designed the experiments, performed the experiments, analyzed the data, prepared figures and/or tables, authored or reviewed drafts of the article, and approved the final draft.
- Yanqiu Liu performed the experiments, analyzed the data, prepared figures and/or tables, and approved the final draft.
- Jiping Cai performed the experiments, analyzed the data, prepared figures and/or tables, and approved the final draft.
- Tianyi Zong performed the experiments, analyzed the data, prepared figures and/or tables, and approved the final draft.
- Ziyin Zhang performed the experiments, analyzed the data, authored or reviewed drafts of the article, and approved the final draft.
- Tianhua Xie performed the experiments, analyzed the data, authored or reviewed drafts of the article, and approved the final draft.
- Tong Mu conceived and designed the experiments, analyzed the data, authored or reviewed drafts of the article, and approved the final draft.

- Meili Wu conceived and designed the experiments, analyzed the data, authored or reviewed drafts of the article, and approved the final draft.
- Qian Yang conceived and designed the experiments, analyzed the data, authored or reviewed drafts of the article, and approved the final draft.
- Yangningzhi Wang conceived and designed the experiments, analyzed the data, authored or reviewed drafts of the article, and approved the final draft.
- Xiaolu Wang conceived and designed the experiments, analyzed the data, prepared figures and/or tables, authored or reviewed drafts of the article, and approved the final draft.
- Yong Yao conceived and designed the experiments, analyzed the data, prepared figures and/or tables, authored or reviewed drafts of the article, and approved the final draft.

### Human Ethics

The following information was supplied relating to ethical approvals (i.e., approving body and any reference numbers):

The ethical approval of this study was obtained from the Ethics Committee of the Affiliated Wuxi People's Hospital of Nanjing Medical University (identifier, KY23013). We obtained written informed consent from every patient.

### DNA Deposition

The following information was supplied regarding the deposition of DNA sequences:

The sequences are available at NCBI: PRJNA971380.

### Data Availability

The raw measurements are available in the Supplementary File.

### Supplemental Information

Supplemental information for this article can be found online at http://dx.doi.org/10.7717/peerj.16569#supplemental-information.

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

# PeerJ

quality of life in the united states. *Ophthalmology and Therapy* **10(1)**:75–87 DOI 10.1007/s40123-020-00318-x.

**Wu L, Li L, Liang Y, Chen X, Mou P, Liu G, Sun X, Qin B, Zhang S, Zhao C. 2021b.** Identification of differentially expressed long non-coding RNAs and mRNAs in orbital adipose/connective tissue of thyroid-associated ophthalmopathy. *Genomics* **113(1 Pt 2)**:440–449 DOI 10.1016/j.ygeno.2020.09.001.

**Wu L, Liang Y, Song N, Wang X, Jiang C, Chen X, Qin B, Sun X, Liu G, Zhao C. 2021a.** Differential expression and alternative splicing of transcripts in orbital adipose/-connective tissue of thyroid-associated ophthalmopathy. *Experimental Biology and Medicine (Maywood)* **246(18)**:1990–2006 DOI 10.1177/15353702211017292.

**Wu L, Zhou R, Diao J, Chen X, Huang J, Xu K, Ling L, Xia W, Liang Y, Liu G, Sun X, Qin B, Zhao C. 2020.** Differentially expressed circular RNAs in orbital adipose/connective tissue from patients with thyroid-associated ophthalmopathy. *Experimental Eye Research* **196**:108036 DOI 10.1016/j.exer.2020.108036.

**Yu G, Wang LG, Han Y, He QY. 2012.** clusterProfiler: an R package for comparing biological themes among gene clusters. *OMICS* **16(5)**:284–287 DOI 10.1089/omi.2011.0118.

**Yue Z, Mou P, Chen S, Tong F, Wei R. 2021.** A novel competing endogenous RNA network associated with the pathogenesis of graves' ophthalmopathy. *Frontiers in Genetics* **12**:795546 DOI 10.3389/fgene.2021.795546.