# Peer review of "Differentially expressed genes in orbital adipose/connective tissue of thyroid-associated orbitopathy"

_PeerJ, doi:10.7717/peerj.16569_

## Round 0.1 · original submission · Major Revisions

Your manuscript has been examined by four reviewers who raised several major and minor points. Please address all the points and pay specific attention to the points raised by reviewer 4. Addressing the concerns of the reviewers will improve the manuscript and make it more comparative for publication in PeerJ.

·

Basic reporting

- Line 28; Abstract: The properties identified by authors are “molecular”, not cellular.
- Abstract: Please specify the number of samples in each group.
- Please Remove sentence between line 68-69- “With the development of transcriptome…”
- Please cite all the tools and methods used in the method section.
- Please provide QC data for the samples at supp data, including sequencing deoth. Also include the results obtained with RNASeqPower.
- Please provide correlation heatmap between PCs with patient demographics (age, gender, and CAS)
- Figure-1 Volcano plot. Please mark and highlight key genes inside the volcano plot.
- Figure1 B. PC1 explains more than 92% of the variation alone. Please provide the Q-Q plot and show box-plots of normalized FPKM values in each gene as supp. data.
- Q-value, adjusted P value, FDR-adjusted P-value and P-values are used. Please use the same terminology across the entire manuscript.
- More results are expected from the functional analysis of the DAS gene.
- Please provide the bubble plot (e.g. REVIGO style) of GO terms.
- Authors must cite the original paper related to the GSE185952 dataset used in this study.
- Please provide venn diagram of the common genes found to be DE between this study and GSE185952.
- PPI network analysis is incomplete. Please provide degree, CC etc information and prioritize DE genes. If possible, please include PPI genes with co-expressed genes with abs(rho) > 0.7

Experimental design

- Sample size per group is very small. Although authors used public data and qRT2PCR for validation, yet those genes that are not found to be validated may lead to incorrect GO/KEGG analysis.

Validity of the findings

no comment

Additional comments

- Please include sample size as study limitation.
- Also discuss those DE genes (include % of these genes) that were not found to be validated in public data and qRT2PCR for validation step.

Reviewer 2 ·

Basic reporting

The authors present a crisp work of finding Differentially expressed genes in orbital adipose/connective tissue of thyroid-associated orbitopathy using bioinformatics plan as well as validating a few of genes using qRT-PCR.
The most of the manuscript presentation is clear in terms of english but I felt it needs a review in terms of citing the literature. For example- line 104 the authors say "We used FeatureCounts..." but did not cite the respective literature.
Line 121 needs to have a re-look wrt formatting (delta-delta Ct method).
Line 260- Gene name should be in capitals.

Experimental design

The Experimental design is clearly stated and for the insight into DEGs it could be used as a potential approach.

Validity of the findings

I have some questions/concerns that I will go through point-wise.

1. PCA analysis (fig1b) showed one TAO sample to be close with controls. Did authors check for the possible reason and if authors checked the analysis removing that particular sample?

2. Authors should also label the samples on the PCA plot.

3. In volcano plot (fig 1) it will be nice if you can label some genes .Specifically those that authors found in validation part as well but try label as many as possible.

4. Since it is unclear from PCA plot which T group sample was closely related to control samples, it is also hard to correlate it to the heatmap for some genes. For example- ALPL (that authors validated also) seems to be relatively downregulated within TAO ( TAO-3 the expression is lower) if compared to two other TAO samples.

Additional comments

The authors presented the work in a very easy going yet crisp manner that I appreciate. They also acknowledged the limitations of the study very clearly. I am not sure though if the data is available on any repository. Kindly clarify on that.

Reviewer 3 ·

Basic reporting

The paper’s is organized; however, some statistical analyses and research conduct are not clear. More details need to be added to support the findings and conclusions. Please see the following comments:

1. The software and statistical tools mentioned in the paper is not properly cited, suggest adding link and citation for each software/tool.

2. There is no legend in the tables and figures. Suggest adding legends to each table and figure to better describe and explain the content. Also, suggest improving the intext reference of the tables/figures, eg. Line 165-174, add reference of each figure in the corresponding text.

3. The raw RNA-seq data is not shared, the author states that “Data will be made available on request.” As the review, I am unable to verify that the data are going to be publicly available.

Experimental design

Some statistical analyses and research conduct are not clear and suggest revising.

1. Table 1 shows 6 samples form control and 5 samples from TAO patients were collected. However, in later analyses, Figure 1, only 3 samples from control and TAO were included. There is no explanation on how and why only 3 samples per group were chosen to include in the analysis.

2. Line 101 states that the RNASeqPower is 0.96 but the assumptions for power calculations is not provided.

3. Table 1, the statistical method to calculate P-value for Age and Gender (Male) is not provided.

4. Line 193, statistical cutoffs were not provided for gene set enrichment analysis.

5. In Figure 5 and Line 234, the statistical method used for p-value calculations is not provided in both figure and texts.

6. Line 136, “P<0.05” supposes to be “q value<0.05” per Line 105 in the method section.

Validity of the findings

Suggest sharing raw RNA-seq data in public database and codes for the analyses in public space for validity of the findings.

Additional comments

No comments

Reviewer 4 ·

Basic reporting

Please see Additional Comments.

Experimental design

Please see Additional Comments.

Validity of the findings

Please see Additional Comments.

Additional comments

The authors combined bioinformatics and qRT-PCR to explore the gene expression patterns of orbital adipose/connective tissue between controls and thyroid-associated orbitopathy patients. Overall, this study is suitable for publication, only if the authors: 1) explain why they focused on orbital adipose/connective tissue, 2) justify why they conducted alternative splicing (DAS) analysis, 3) pinpoint how this research advances our understanding of thyroid-associated orbitopathy (that is, the novelty of the current research compared to other similar articles), and 4) address the following issues:

1. Throughout the manuscript, it seems better to use Grammarly (https://www.grammarly.com/) to check & correct potential grammatical errors or typos.

2. In all FIGURES, it would be clear and more readable to expand on figure legends by explaining the meanings of colors, groups, lines, and abbreviations. These revisions would greatly help readers, who do not specialize in bioinformatics, to understand the results and their implications easily and efficiently.

3. In ABSTRACT:
3.1 In Background, it seems better to change "Therefore, to comprehensively understand the molecular mechanisms underlying orbital adipogenesis in TAO" into "To comprehensively understand the molecular mechanisms ...", which would be more concise and cohesive (that is, sentences are closely connected).
3.2 In Background, it would be more informative and clearer to mention the research/knowledge gap — the question that was not answered by previous studies (about orbital adipose/connective tissue of thyroid-associated orbitopathy) but is being addressed by the current manuscript. (a pattern like PMID: 34715879, PMID: 34384362, PMID: 35965679, and PMID: 34537192) To be more specific, comparing the current study with previous research would help the authors summarize the novelty of this study and pinpoint the research/knowledge gap.
3.3 In Background, it would be more informative and cohesive to explain why it is important to analyze the "adipose/connective tissue" (, perhaps because of its importance in the pathogenesis of thyroid-associated orbitopathy?).
3.4 In Results, it would be more informative to mention the number of up- and down-regulated DEGs.
3.5 In Results, it would be clearer to mention whether the following results were derived from sequencing or qPCR: "They include ceruloplasmin isoform x3 (CP), alkaline tissue-nonspecific isozyme isoform x1 (ALPL), and angiotensinogen (AGT), which were overrepresented by 2.27- to 6.40-fold. Meanwhile, protein mab-21-like 1 (MAB21L2), phosphoinositide 3-kinase gamma-subunit (PIK3C2G), and clavesin-2 (CLVS2) decreased by 2.6%- to 32.8%. R-spondin 1 (RSPO1), which is related to oogonia differentiation and developmental angiogenesis, was significantly downregulated in the orbital muscle tissues of patients with TAO compared with the control groups (P=0.024)."
3.6 In Conclusions, it would be more rigorous and accurate to rewrite "Our results suggest that there are intrinsic genetic differences in orbital adiposeconnective tissues derived from TAO patients", in which "intrinsic genetic differences" did not seem to be supported by the data. Because the data between patients and controls characterized the genetic differences that did not seem to be "intrinsic" but rather between patients and controls.
3.7 In Conclusions, it would be clearer to rewrite "The upregulation of the inflammatory response in orbital fat of TAO may be consistent with the clinical phenotype" by mentioning what is "the clinical phenotype".
3.8 In Conclusions, please justify why the authors mentioned "MAB21L2, PIK3C2G, and CLVS2" but ignored other genes that were mentioned in Results. Otherwise, please delete the sentence "Downregulation of MAB21L2, PIK3C2G, and CLVS2 in TAO tissue demonstrates dysregulation of differentiation, oxidative stress, and developmental pathways", which did not seem to be supported by convincing data from experiments.

4. In INTRODUCTION:
4.1 In Paragraph 1, it seems better to change "Expansion of the orbital tissue volume (extraocular adipose and muscle tissue) surrounding the eye, may cause orbital congestion, significant exophthalmos, compressive neuropathy, and even lead to vision loss reducing a serious decline in quality of life" into "... vision loss causing a serious decline in quality of life", which appears to be more accurate.
4.2 In Paragraph 2, it would be more informative and connected with TITLE to rewrite this paragraph by introducing the orbital adipose/connective tissue-related molecular mechanisms (genes) behind thyroid-associated orbitopathy. This introduction would better justify why the authors carried out sequencing in "orbital adipose/connective tissue of thyroid-associated orbitopathy".
4.3 In Paragraph 3, it seems better to change "Then, Gene Ontology (GO), Kyoto Encyclopedia of Genes and Genomes (KEGG) and Gene set enrichment analysis (GSEA) pathway analyses were obtained to screen out significantly enriched functions" into "... and Gene set enrichment analysis (GSEA) pathway analyses were obtained to predict functions of these DEGs", which would be more accurate and clearer.
4.4 In Paragraph 4, it seems better to change "Furthermore, we confirmed the result by quantitative real-time polymerase chain reaction (qRT2PCR)" into "The expression patterns of some DEGs were confirmed by qRT-PCR", which would be more cohesive and informative.

5. In MATERIALS & METHODS:
5.1 In "Differentially expressed genes (DEGs) and differential alternative splicing (DAS) analysis", it would be more informative to explain BOTH why the authors identified "splicing events" AND why the authors used "the software rMATS".
5.2 In "Functional enrichment analysis", it would be more rigorous to cite references or websites when mentioning "the software topGO" and "GSEA and KEGG Orthology Based Annotation System (KOBAS) v3.0".
5.3 In "RNA quantification", it would be more reproducible and rigorous to mention the procedure for RNA extraction, as well as reagents that were used during the extraction process.

6. In RESULTS:
6.1 It would be clearer to end each paragraph in RESULTS with one sentence: "Together, these results suggest that ..." (a pattern like PMID: 34715879, PMID: 34384362, PMID: 35965679, and PMID: 34537192), summarizing a paragraph AND highlighting the implications of all results in the paragraph.
6.2 In "Differential alternative splicing (DAS) gene analysis", "Alternative splicing (AS) refers to the process of selectively removing or retaining exons/introns during the initial transcription of DNA into RNA and further processing into mature mRNA, resulting in multiple transcripts of a gene" did not seem to be sufficient for justifying why the authors carried out the DAS analysis. It would be more informative to summarize the importance of DAS in thyroid-associated orbitopathy (or similar diseases), perhaps based on references.
6.3 In "Differential alternative splicing (DAS) gene analysis", it would be more informative and valuable to rewrite this section by extracting more meaningful data (or implications), because "The numbers of A3SS, A5SS, MXE, RI, and SE events" did not seem to imply anything. Compared with "the numbers of ... events", it appears to be more important to analyze what genes are influenced by these events.

7. In SUPPLEMENTAL MATERIALs, it seems better to add English versions of "human participants consent form", because this revision would help international readers to understand.

---

## Round 0.2 · Minor Revisions

Reviewers
Although you have adequately addressed most of the reviewer’s points, minor revision is still required. Please address the below two points and re-submit a revised version of the manuscript.

Specifically;

1. Reviewer 1; recommended grammatical corrections for the manuscript.

2. Reviewer 2; questioned that “Point- PCA analysis (fig1b) showed one TAO sample to be close with controls. Did the authors check for the possible reason and if authors checked the analysis removing that particular sample”?

Ziarih Hawi

·

Basic reporting

no comment

Experimental design

no comment

Validity of the findings

no comment

Additional comments

no comment

Reviewer 2 ·

Basic reporting

See the comments

Experimental design

see the comments below

Validity of the findings

See the comments below

Additional comments

I appreciate that the authors proposed their justification on most of my concerns but I did not quite understand specifically one point where authors justify one of my points-

Point- PCA analysis (fig1b) showed one TAO sample to be close with controls. Did authors check for the possible reason and if authors checked the analysis removing that particular sample?

Response: We very appreciated the reviewer’s concern. We regret that we did not remove that particular sample, and we will increase the sample size for validation and to perform functional verification.

What do authors mean by saying that they regret not removing that particular sample and WILL increase the sample size for validation. Are they talking about increasing the sample size for this manuscript? I am not sure that authors justified that.

Other than this point, I am satisfied with the manuscript progress.

Reviewer 3 ·

Basic reporting

no comment

Experimental design

no comment

Validity of the findings

no comment

Additional comments

no comment

Reviewer 4 ·

Basic reporting

Please see Additional Comments.

Experimental design

Please see Additional Comments.

Validity of the findings

Please see Additional Comments.

Additional comments

Thank the authors for their efforts to respond to all of my comments. The current version would be suitable for publication.

---

## Round 0.3 · Minor Revisions

You have adequately addressed the concerns of the reviewers and have made significant changes towards making the paper suitable for publication in PeerJ. However we notice that this work is very similar to other published works,

Please explain what distinguishes this study from other similar studies, and cite if appropriate:

- https://doi.org/10.1016/j.ygeno.2020.09.001
- https://doi.org/10.3389/fendo.2022.1001349
- https://doi.org/10.3389/fcell.2021.716871
- https://doi.org/https://doi.org/10.1167/iovs.15-17185

There appear to be more - please do a thorough literature search.

---

## Round 0.4 · accepted · Accept

Thank you for addressing the concerns of the reviewers and making significant changes to the manuscript. Your manuscript is ready for publication.